# Prevalence and change in alcohol consumption in older adults over time, assessed with self-report and Phosphatidylethanol 16:0/18:1 —The HUNT Study

Kjerstin Tevik[1,2]*, Ragnhild Bergene Skråstad[3,4], Jūratė Šaltytė Benth[5,6], Geir Selbæk[1,7,8], Sverre Bergh[1,9], Rannveig Sakshaug Eldholm[10,11], Steinar Krokstad[12,13], Anne-Sofie Helvik[1,2]

1 Norwegian National Centre for Ageing and Health, Vestfold Hospital Trust, Tønsberg, Norway, 2 Department of Public Health and Nursing, Faculty of Medicine and Health Sciences, Norwegian University of Science and Technology (NTNU), Trondheim, Norway, 3 Department of Clinical Pharmacology, St. Olavs Hospital, Trondheim University Hospital, Trondheim, Norway, 4 Department of Clinical and Molecular Medicine, Faculty of Medicine and Health Sciences, Norwegian University of Science and Technology (NTNU), Trondheim, Norway, 5 Institute of Clinical Medicine, Campus Ahus, University of Oslo, Oslo, Norway, 6 Health Services Research Unit, Akershus University Hospital, Lørenskog, Norway, 7 Department of Geriatric Medicine, Oslo University Hospital, Oslo, Norway, 8 Institute of Clinical Medicine, Faculty of Medicine, University of Oslo, Oslo, Norway, 9 Research Centre for Age-Related Functional Decline and Disease, Innlandet Hospital Trust, Ottestad, Norway, 10 Department of Geriatrics, St. Olavs Hospital, Trondheim University Hospital, Trondheim, Norway, 11 Department of Neuromedicine and Movement Science, Faculty of Medicine and Health Sciences, Norwegian University of Science and Technology (NTNU), Trondheim, Norway, 12 HUNT Research Centre, Department of Public Health and Nursing, Faculty of Medicine and Health Sciences, Norwegian University of Science and Technology (NTNU), Trondheim, Norway, 13 Levanger Hospital, Nord-Trøndelag Hospital Trust, Levanger, Norway

* kjerstin.e.tevik@ntnu.no

**Data Availability Statement:** Due to restrictions imposed by the HUNT Research Centre (in accordance with the Norwegian Data Inspectorate), data cannot be made publicly available. Data are

## Abstract

### Background

Changes in alcohol consumption may affect older adults' health. We examined prevalence and changes in the alcohol consumption of older women and men (≥65 years) in Norway over a 24-year period.

### Methods

Data from three population-based health surveys (The Trøndelag Health Study—HUNT2 1995–97, HUNT3 2006–08, HUNT4 2017–19) were used. Alcohol consumption was measured using self-reported measures and an objective measure of alcohol consumption (Phosphatidylethanol 16:0/18:1, PEth). Self-reported lifetime abstinence, former drinking, current drinking, frequent drinking (≥4 times/week), and risk drinking (≥8 units/week) were measured. The PEth concentrations were stratified: <0.03 μmol/l (abstinence/very low level of alcohol consumption); >0.06 μmol/l (indicating >1 unit/day); >0.10 μmol/l (indicating >3 units/day), and >0.30 μmol/l (heavy alcohol consumption).

currently stored in the HUNT databank, and there are restrictions in place for the handling of HUNT data files. Data used from the HUNT Study in research projects will be made available on request to the HUNT Data Access Committee (kontakt@hunt.ntnu.no). The HUNT data access information (available here: http://www.ntnu.edu/hunt/data) describes in detail the policy regarding data availability.

**Funding:** This project was funded by the Swedish STAFF foundation (Stiftelsen Ansvar för fremtiden) (https://ansvarforframtiden.se/) through the Norwegian organization ACTIS (https://actis.no/). In addition, the project has been funded partly by the Norwegian National Centre for Ageing and Health (Ageing and Health), Vestfold Hospital Trust (https://www.aldringoghelse.no/english/). KT received the funding from the STAFF foundation and from Ageing and Health. This project has also been funded by the Clinic of Laboratory Medicine at the Department of Clinical Pharmacology at St. Olavs University Hospital in Trondheim (Norway) (PEth analyses) (https://www.stolav.no/) and the Norwegian DAM foundation (https://dam.no/) through the Norwegian non-profit organization 'Av og til' (https://avogtil.no/). RBS received the funding from St. Olavs University Hospital and the DAM foundation. The funders had no role in study design, data collection and analysis, decision to publish, or preparation of the manuscript.

**Competing interests:** The authors have declared that no competing interest exist.

## Results

In HUNT4, the prevalence of self-reported lifetime abstinence, frequent drinking, and risk drinking was 5.2%, 4.4%, and 5.6%, respectively, while prevalence of PEth <0.03 µmol/l was 68.1% and PEth >0.06 µmol/l was 21.2%. Over the course of the three surveys, the prevalence of self-reported lifetime abstinence decreased, while the prevalence of frequent drinking and risk drinking increased. Men were less often abstainers and more often frequent and risky drinkers than women in all three surveys. Gender differences for abstinence and current drinking reduced with time. From HUNT3 to HUNT4, the prevalence of PEth <0.03 µmol/l decreased, while the prevalence of PEth >0.06 µmol/l increased. Men compared to women, had less often PEth <0.03 µmol/l and more often PEth >0.06 and >0.10 µmol/l in HUNT3 and HUNT4. Women and men ≥75 years were just as likely to have PEth >0.30 µmol/l in HUNT4. The gender differences in PEth concentrations were reduced in HUNT4 among those aged 70–74 years or ≥75 years.

## Conclusion

Alcohol consumption has increased among Norwegian older adults over a 24-year period, but at a slower pace during the last decade.

## Introduction

The Western population is rapidly aging [1, 2]. In an aging population, it is important to have knowledge about prevalence and change in alcohol consumption, as older adults are more sensitive to alcohol than younger adults [3, 4]. Even though alcohol consumption tends to decline in older age [5, 6], drinking alcohol is common among older adults [7–9], and about 80% drink alcohol at least once per year [7, 10]. Further, the prevalence of frequent drinking (≥4 times per week) (16–39%) and risk drinking (≥8 units of alcohol per week) (21–47%) in older adults has been shown to be high in several studies [6, 8, 10, 11].

In Western countries [12–16], including Norway [5, 7, 9, 17], older adults have also changed their drinking patterns in the last three decades. The proportion of older adults entirely abstaining from alcohol are reduced [9, 13, 17]. Furthermore, current drinking [7, 13, 15] and risk drinking [7, 9, 12, 14, 15] have increased among older adults in the USA, Canada, and in several European countries (Norway, Sweden, Germany) [7, 9, 12–16]. In other European countries (Italy, Finland, Spain) [18, 19] and in Australia [20], risk drinking has been quite stable or decreased in recent decades among older adults. However, it is difficult to compare the true prevalence of and changes in alcohol consumption due to the huge variations and no international consensus in definitions and assessment methods used in epidemiologic alcohol studies in older adults [4].

The high and increasing prevalence of elevated alcohol consumption (frequent and risk drinking) among older adults in several countries in recent decades is of concern and may have important public health implications, as alcohol consumption is a major risk factor for injury, disability, burden of disease, and mortality [21–23]. Compared with younger adults, older adults are more susceptible to adverse health effects of alcohol due to reduced tolerance and increased sensitivity to alcohol [3, 4]. In older adults, the ability to metabolize alcohol declines and the body composition changes with decreased body water and increased body fat, leading to higher blood alcohol concentration and prolonged effects of alcohol [3, 4]. In

addition, comorbidity and polypharmacy increase the risk of negative effects of alcohol in older adults [3, 24]. Alcohol consumption in older adults may lead to injuries, falls, and fractures [25, 26], and may increase the risk for undernutrition, frailty, mental health problems (depression and anxiety) [27–30], and alcohol and drug interaction [31, 32]. Elevated alcohol consumption is associated with liver disease [22, 33], cardiovascular diseases [34, 35], cancer [22, 36], dementia [37], and increased mortality [22, 34, 36], both in the general population and among older adults.

Older men consume more alcohol than older women, including more frequent drinking and more risk drinking [5, 9, 11, 38]. However, during recent decades, the increase in alcohol consumption has been particularly pronounced among older women, and the gender differences in alcohol consumption among older adults have reduced [9, 10, 12, 13]. The gender convergence is of concern as women compared to men are susceptible to more severe brain and other organ damage following elevated alcohol consumption (i.e., episodic or chronic alcohol abuse) [39, 40]. The increased risk among women may be due to gender differences in alcohol metabolism and lower total body water, leading to higher alcohol concentration in blood after drinking equivalent volumes of alcohol and gender differences in genetic and neurobiological factors [39–41].

From a public health perspective, it is important to examine how the prevalence of alcohol consumption might be changing in older women and men, given the potential for adverse health effects, even at low level of alcohol consumption [21, 22]. This information can serve as a basis for future health care planning, resource allocation, and public health efforts to reduce alcohol consumption [12].

Internationally, we have limited knowledge about alcohol consumption in a total population of older adults, and in Norway there are few recent studies that have examined prevalence, change in prevalence, and gender differences in alcohol consumption among older adults [7, 9]. Previous studies have only used self-reported alcohol consumption to define different drinking patterns [7, 9], which is susceptible to underreporting, especially among heavy drinkers [42, 43]. Worldwide, we are lacking data on objective measures of alcohol consumption in older adults [44]. Biological markers of alcohol intake, so-called alcohol markers, provide an opportunity to measure alcohol consumption in an objective manner. Of the currently available alcohol markers, Phosphatidylethanol 16:0/18:1 (PEth), measured in whole blood, seems to be among the most sensitive and specific [45, 46].

The first aim was to study the prevalence and changes in alcohol consumption among older adults (≥65 years) by gender and age, using self-report measures in three independent large cross-sectional population-based health surveys from the Trøndelag Health Study (HUNT2 1995–97, HUNT3 2006–08, and HUNT4 2017–19) [47], and using an objective measure of alcohol consumption (PEth) in two of the HUNT surveys (HUNT3 2006–08 and HUNT4 2017–19). Secondly, we wanted to study gender differences in these patterns, stratified by age categories within each HUNT survey and between the HUNT surveys by the use of self-reported and objective measures of alcohol intake.

## Material and methods

### Study design

We used data from three consecutive cross-sectional population-based health surveys conducted in the region of Nord-Trøndelag in Norway (HUNT2, HUNT3, and HUNT4) to examine the prevalence and changes in alcohol consumption among older adults (≥65 years) [47].

## Study setting, data sources, and participants

The Trøndelag Health Study (HUNT) is a population-based cohort study of the adult population that is conducted in Trøndelag county in Mid-Norway. Surveys of the citizens of Nord-Trøndelag have been conducted every decade since 1984–86 [48]. All residents in Nord-Trøndelag aged 20 years or older are invited to participate in each HUNT survey. The total participation rate for HUNT2, HUNT3, and HUNT4 was 69.5% (65,237 of 93,898 invited), 54.1% (50,807 of 93,860 invited), and 54.0% (56,042 of 103,800 invited), respectively [48–50]. Nord-Trøndelag is considered to be fairly representative of Norway regarding geography, industry, age distribution, morbidity, and mortality [49]. However, the habitants in Nord-Trøndelag have a slightly lower educational level and lower income compared to Norway as a whole [49, 51], and Nord-Trøndelag does not have any large cities and has lower numbers of immigrants compared to Norway as a whole [48].

This study used data only from non-institutionalized older adults (≥65 years) [48–50]. S1 Table describes the participation rates for individuals aged 60 years and older in HUNT2, HUNT3, and HUNT4 which declined by increasing age group [48–50]. A non-participation study of all participants after HUNT2 showed only minor potential non-participation bias [52], while non-participants in HUNT3 had lower socioeconomic status, poorer health, and a higher prevalence of chronic diseases and mental distress compared to participants [52]. However, there was no difference in drinking ≥2–3 times per week between participating and non-participating men older than 60 years, or between participating and non-participating women aged 60–79 years in HUNT3. Participating women older than 80 years drank more often (≥2–3 times per week) than non-participating women in HUNT3 [52]. Compared to all participants in HUNT4, a higher proportion of non-participants had poorer self-rated health and a lower proportion consumed alcohol ≥2–3 times per week. Home nursing and smoking were more common among non-participants in HUNT4 than among participants [48].

Each HUNT survey performed in Nord-Trøndelag consists of two self-report questionnaires (Q1 and Q2), interviews and clinical examinations at an examination station, laboratory measurements, and taking and storage of biological samples [48–50]. The HUNT questionnaires can be found on the HUNT webpage: https://www.ntnu.edu/hunt/data/que. Full details of the HUNT Study have been described elsewhere [48–50].

Data on income was provided by Statistics Norway (SSB) [53], and was linked to HUNT2, HUNT3, and HUNT4 data for the participants. In the analyses we used income after taxes as a covariate. Values of 0- or negative income for the year of participation were replaced by average of the remaining two values (or one value if only single value available). 0-income for all three years were replaced with missing.

**Participants.** This study included individuals who were ≥65 years when participating, and who had answered the first questionnaire (Q1) and the question about drinking frequency and/or volume of alcohol consumption in HUNT2, HUNT3, and/or HUNT4.

## Measures

**Alcohol consumption.** The three HUNT surveys (HUNT2, HUNT3, and HUNT4) included data from self-report questions about drinking-frequency and volume of alcohol consumption [54]. Table 1 provides more detailed information about the questions used to assess alcohol consumption in HUNT2, HUNT3, and HUNT4. The recall period, the type of questions, and the wording of the questions for alcohol consumption varied to some extent across the surveys. The recall period for drinking frequency in HUNT2 was the past month, while in HUNT3 and HUNT4 it was the past year. Self-report data was used to examine the prevalence

**Table 1. Questions used to assess alcohol consumption in HUNT2, HUNT3, and HUNT4.**

| Assessment of different drinking patterns | Questions in HUNT2 1995–97 | Questions in HUNT3 2006–08 | Questions in HUNT4 2017–19 |
|---|---|---|---|
| Assessment of drinking frequency | Q1: Concerning alcohol: Do you entirely abstain from alcohol (yes or no)<br>Q1: How many times a month do you normally drink alcohol?<br>(Number of times____)<br>(Do not include low-alcohol beer.<br>Put 0 if less than once a month.) | Q1: About how often in the last 12 months did you drink alcohol? (do not include low-alcohol beer)<br>4–7 times a week<br>2–3 times a week<br>About once a week<br>2–3 times a month<br>About once a month<br>A few times a year<br>Not at all the last year<br>Never consumed alcohol | Q1: About how often in the last 12 months did you drink alcohol (do not include low-alcohol beer)<br>Not at all last 12 months<br>Once a month or less<br>2–4 times a month<br>2–3 times a week<br>4 or more times a week<br>I have never consumed alcohol |
| Assessment of volume of alcohol consumption | Q1: How many glasses of beer, wine, or spirits do you usually drink in the course of two weeks? (Do not include low-alcohol beer. Put 0 if less than once a month.)<br>Beer (Number of glasses ____)<br>Wine (Number of glasses __)<br>Spirits (Number of glasses __) | Q1: How many glasses of beer, wine, or spirits do you usually drink in the course of two weeks: (do not include low-alcohol beer, write 0 if you do not drink alcohol)<br>Beer (Number of glasses ____)<br>Wine (Number of glasses __)<br>Spirits (Number of glasses __) | Q1: How many glasses of beer, wine, or spirits do you usually drink in the course of two weeks: (do not include low-alcohol beer, write 0 if you do not drink alcohol)<br>Beer (Number of glasses ____)<br>Wine (Number of glasses __)<br>Spirits (Number of glasses __) |
| Assessment of PEth | No information | Measures of PEth in whole blood in a subsample | Measures of PEth in whole blood in a subsample |

Abbreviations: HUNT = Trøndelag Health Study; PEth = Phosphatidylethanol 16:0/18:1; Q1 = The first questionnaire.

and changes in lifetime abstaining, former drinking, current drinking, and elevated alcohol consumption (frequent drinking and risk drinking) (Table 2).

PEth was measured in whole blood in a subsample in HUNT3 and HUNT4 [55, 56]. In HUNT3, PEth was analyzed in stored blood from the HUNT-biobank and material was only available from a subsample. For practical reasons in HUNT4, the collection of blood samples for PEth-analysis did not start until approximately halfway through the study period, and thus PEth-analysis is only available in a subsample. In both cases, the subsample was independent of whether the participants reported to have a high or a low level of alcohol consumption. The prevalence and changes of different levels of alcohol consumption were examined using PEth analyses (Tables 1 and 2).

*Lifetime abstention.* In HUNT2, the participants were asked: "Do you entirely abstain from alcohol?". Those who reported "yes" to this question, were defined as lifetime abstainers. In HUNT3 and HUNT4 the participants were asked how often during the last 12 months they drank alcohol. Those who reported "never consumed alcohol" were defined as lifetime abstainers (Table 2) [57].

*Former drinking.* In HUNT3 and HUNT4, those who reported "not consumed alcohol in past year" in the drinking frequency questionnaire, were defined as former drinkers (Table 2) [57]. HUNT2 has no information about former drinkers.

*Current drinking.* In HUNT2, the participants were asked how many times a month they drank alcohol. Those who reported drinking once per month or more were defined as current drinkers. In HUNT3, those who reported "drinking few times per year" or a higher drinking frequency category, were defined as current drinkers. In HUNT4, those who reported drinking "once per month or less" or responded a higher drinking frequency category, were defined as current drinkers (Table 2).

**Table 2. Definition of abstainers, former drinkers, current drinkers, elevated alcohol consumption, and alcohol consumption assessed by PEth.**

| Drinking patterns[1] | Definition HUNT2 (1995–97) | Definition HUNT3 (2006–08) | Definition HUNT4 (2017–19) |
|---|---|---|---|
| **Abstainers and current drinkers** | | | |
| Lifetime abstainer | Total abstinence from alcohol | Never consumed alcohol in lifetime | Never consumed alcohol in lifetime |
| Former drinker | No information | Not consumed alcohol during the past year | Not consumed alcohol during the past year |
| Current drinker | Consumed alcohol ≥once per month | Consumed alcohol ≥few times per year | Consumed alcohol ≥few times per year |
| **Elevated alcohol consumption** | | | |
| Frequent drinking | Drinking alcohol ≥4 times per week | Drinking alcohol ≥4 times per week | Drinking alcohol ≥4 times per week |
| Risk drinking | Drinking ≥8 units of alcohol per week | Drinking ≥8 units of alcohol per week | Drinking ≥8 units of alcohol per week |
| **Different levels of alcohol consumption measured by PEth** | | | |
| Abstinence or a very low level of alcohol consumption | No data | <0.03 µmol/l | <0.03 µmol/l |
| >1 alcohol unit/day | No data | >0.06 µmol/l | >0.06 µmol/l |
| >3 alcohol units/day | No data | >0.1 µmol/l | >0.1 µmol/l |
| Heavy alcohol consumption | No data | >0.3 µmol/l | >0.3 µmol/l |

Abbreviations: HUNT = Trøndelag Health Study; PEth = Phosphatidylethanol 16:0/18:1.

[1]The HUNT questionnaires used to define and describe different self-reported drinking patterns can be found in the following link: https://www.ntnu.edu/hunt/data/que

*Frequent drinking.* In HUNT2, participants who reported drinking 16 times or more per month were considered equivalent to consuming alcohol 4 times or more per week, and were defined as frequent drinkers [5]. In HUNT3 and HUNT4, participants who reported drinking "4–7 times per week" or "4 or more times per week" in the drinking frequency questionnaire, were defined as frequent drinkers (Table 2). This cut-off is used in other comparable studies [5, 58, 59].

*Risk drinking.* In HUNT2, HUNT3, and HUNT4, the participants reported the number of glasses of beer, wine, or spirits they usually consumed in two weeks. In our study, this was converted to total number of glasses of beer, wine, or spirits consumed per week. One glass of beer, wine, or spirits was classified to be equivalent to one unit of alcohol. In Norway, one unit of alcohol is defined as ca. 12 grams (g) of pure alcohol [9, 60], corresponding to approximately one unit of beer (0.33 L) with 4.5% alcohol by volume (ABV), one unit of wine (0.125 L) with 12% ABV, and one unit of spirits (0.04 L) with 40% ABV [60, 61]. Risk drinking was defined as consuming ≥8 units of alcohol per week for both women and men (Table 2). This definition is commonly used in epidemiologic studies assessing risk drinking in older adults [4], and is in line with alcohol guidelines for older adults in the USA [62]. In Norway, consuming 8 units of alcohol per week is equal to approximately 96 g of pure alcohol (8 units of alcohol x 12 g of pure alcohol), which corresponds approximately to the threshold values defined by Wood et al. [21] as being associated with increased health risk (≥100 g per week).

*PEth analyses.* PEth was measured/analyzed in HUNT3 (≥65 years: n = 6,068) and HUNT4 (≥65 years: n = 7,290). The PEth concentration reflects the size of the alcohol intake during the last 2–4 weeks before sampling [63, 64]. Blood samples were drawn into 3 ml EDTA tubes, placed in refrigerated storage overnight, and then frozen and stored at –80 C until analysis. PEth was analyzed in whole blood with a validated ultra-performance liquid chromatography tandem mass spectrometry (UPLC®-MSMS) method published elsewhere, with a

quantification range of 0.030–4.00 µmol/l [55]. For calculations from µmol/l to ng/ml, we used a conversion-factor of 703. A PEth concentration below 20 ng/ml (0.028 µmol/l) has been proposed as compatible with abstinence or very low alcohol consumption. A PEth concentration of 200 ng/ml (0.28 µmol/l) has been proposed as a threshold for "heavy consumption", corresponding to an average consumption of 60 g or more of pure ethanol on a single drinking day over a prolonged duration for men and 40 g for women [64, 65]. However, these cut-off values do not take into account that, in some situations and populations, including among older adults, a lower alcohol consumption can be harmful to health. A recently published study based on the HUNT4 material advocates differentiating between the cut-off value used based on the alcohol consumption measured in the average number of units consumed per day, and the desired level of specificity and sensitivity [56]. Based on results from previous studies [56, 64, 65], PEth concentrations were stratified/grouped in the present study as follows: 1) PEth <0.03 µmol/l = abstinence or a very low level of alcohol consumption [64, 65]; 2) PEth >0.06 µmol/l = consumption of more than one alcohol unit/day [56]; 3) PEth >0.10 µmol/l = consumption of more than 3 alcohol units/day [56]; 4) PEth >0.30 µmol/l = heavy alcohol consumption/heavy drinking [64] (Table 2).

**Demographic and socioeconomic variables.** Demographic and socioeconomic variables known to be associated with alcohol consumption were included, such as gender, age at the time of survey completion, level of education (up to 10 years of education, vocational and general education, and college and university), income after taxes (mean, median, first and third quartile), marital status (living with spouse or partner versus not), smoking (never smoked, previously smoked, and smoker), and living place (urban versus rural living) [5, 14, 66, 67]. Nord-Trøndelag consists of 23 municipalities. Five have status as cities and were defined as urban areas in the HUNT Study [68]. Age was categorized into three groups (65–69 years, 70–74 years, and 75 years or older). With the exception of the information about income from SSB, all information was collected from HUNT2, HUNT3, and HUNT4. All HUNT data except PEth was based on self-report.

## Ethics

All participants signed an informed written consent allowing the use of their data for future medical research [48–50]. This consent allows their data to be linked to other health and administrative registries in Norway, such as SSB [48]. SSB merged the data from HUNT2, HUNT3, HUNT4, and SSB [53]. The merging of HUNT and SSB data is made possible through the unique Norwegian 11-digit personal identification (ID) numbers [50]. To ensure anonymity according to Norwegian regulations for merging data from different health registers, all personal identification data (names and personal ID-numbers) was removed from the data files.

HUNT research is carried out in accordance with the Regional Committee of Medical and Health Research Ethics (REC), the Norwegian Data Inspectorate Authority, and applicable law [47]. The present study is approved by REC (reference number 407997) and the Norwegian Social Science Data Services (reference number 419689).

## Statistics

Characteristics for the entire sample, as well as stratified by gender (women and men), age group (65–69, 70–74, and ≥75), and survey (HUNT2, HUNT3, and HUNT4) were presented as means and standard deviations (SDs) for continuous data, and frequencies and percentages for categorical data.

Prevalence (frequency and percentage) in self-reported alcohol consumption (lifetime abstainer, former drinker, current drinker, and elevated alcohol consumption [frequent drinking and risk drinking]) and alcohol consumption assessed by PEth (dichotomized as <0.03 vs. ≥0.03, >0.06 vs. ≤0.06, >0.10 vs. ≤0.10, and >0.30 vs. ≤0.30 μmol/l) were presented for the entire sample, and stratified by gender, age group, and survey.

A logistic regression model was estimated to assess gender differences within each HUNT survey, as well as gender differences across the health surveys, all stratified by age. The model included dummies for gender, age, and HUNT survey, and all two and three-way interactions. The results were presented as odds ratios (ORs) for gender differences (men vs. women), and as ORs for differences between the surveys regarding gender differences (all pairwise comparisons) with corresponding 95% confidence intervals (CIs). Crude odds for different patterns of alcohol consumption within strata defined by gender and age at each survey were illustrated graphically. Crude ORs and ORs adjusted for marital status and income after taxes are presented in the tables. Only statistically significant adjusted results regarding gender differences within each HUNT survey and across the HUNT survey will be presented. Overall changes in alcohol consumption assessed with self-report and PEth across the surveys will be presented only descriptively.

As some participants were included in multiple surveys, within-participant correlations might have been present in data. Such correlations were assessed using the intra-class correlation coefficient, and were adjusted for by including random effects for participants in the regression model whenever necessary.

All statistical analyses were performed in STATA v.17.

## Results

Table 3 shows the basic characteristics of the participants (≥65 years) in HUNT2 (N = 14,090; 54.3% women), HUNT3 (N = 11,903; 53.7% women), and HUNT4 (N = 17,124; 52.6% women). The mean age (SD) was 74.0 (6.1) years in HUNT2, 73.9 (6.4) in HUNT3, and 74.1 (6.7) in HUNT4.

In the subsample with measured PEth at HUNT3 (n = 6,068, 52% women) and HUNT4 (n = 7,290, 52.5% women) the mean age (SD) was 73.6 (6.2) in HUNT3 (S2 Table), and 74.1 (6.7) in HUNT4 (S3 Table). Participants with measured PEth in HUNT3 were more often men, were younger, had lower income, and lived in urban areas more often than participants without measured PEth (S2 Table). In HUNT4, the basic characteristics between participants with and without measured PEth were almost at the same level. The exception was that a lower proportion of participants with measured PEth lived in urban areas compared to those without measured PEth (S3 Table).

### Prevalence and changes in self-reported alcohol consumption

Information about the prevalence of self-reported alcohol consumption in each HUNT survey is found in Table 4. In HUNT4, the prevalence of lifetime abstinence, current drinking, frequent drinking, and risk drinking was 5.2%, 80.6%, 4.4%, and 5.6%, respectively. Fig 1a–1e illustrates graphically the crude odds for each self-reported patterns of alcohol consumption among women and men stratified by age (65–69, 70–74, and ≥75 years) in HUNT2, HUNT3, and HUNT4.

The total prevalence of lifetime abstinence decreased in both genders, and decreased more from HUNT2 to HUNT3 than from HUNT3 to HUNT4. Along with the decline in total abstinence, the prevalence of current drinking increased among both women and men from HUNT2 to HUNT3, while the prevalence decreased slightly from HUNT3 to HUNT4. The

**Table 3. Overall sample characteristics of women and men ≥65 years at HUNT2 (1995–97), HUNT3 (2006–08), and HUNT4 (2017–19), stratified by age.** Numbers are frequencies and percentages (%) unless stated otherwise.

| Characteristic | HUNT2 1995–1997 | | | HUNT3 2006–2008 | | | HUNT4 2017–2019 | | |
|---|---|---|---|---|---|---|---|---|---|
| | Women (n = 7,644) | Men (n = 6,446) | Total (n = 14,090) | Women (n = 6,388) | Men (n = 5,515) | Total (n = 11,903) | Women (n = 9,000) | Men (n = 8,124) | Total (n = 17,124) |
| Age, years | | | | | | | | | |
| Mean (SD) | 74.4 (6.2) | 73.6 (6.0) | 74.0 (6.1) | 74.1 (6.6) | 73.6 (6.2) | 73.9 (6.4) | 74.5 (7.0) | 73.8 (6.5) | 74.1 (6.7) |
| Age groups | | | | | | | | | |
| 65–69 | 2,220 (29.0) | 2,075 (32.2) | 4,295 (30.5) | 2,128 (33.3) | 1,966 (35.6) | 4,094 (34.4) | 2,819 (31.3) | 2,751 (33.9) | 5,570 (32.5) |
| 70–74 | 2,146 (28.1) | 1,976 (30.7) | 4,122 (29.3) | 1,611 (25.2) | 1,506 (27.3) | 3,117 (26.2) | 2,603 (28.9) | 2,441 (30.0) | 5,044 (29.5) |
| ≥75 | 3,278 (42.9) | 2,395 (37.2) | 5,673 (40.3) | 2,649 (41.5) | 2,043 (37.0) | 4,692 (39.4) | 3,578 (39.8) | 2,932 (36.1) | 6,510 (38.0) |
| Education[a] | | | | | | | | | |
| Up to 10 years | 5,199 (80.4) | 3,486 (61.1) | 8,685 (71.4) | | | | 2,828 (31.9) | 1,547 (19.2) | 4,375 (25.8) |
| Vocational and general | 946 (14.6) | 1,717 (30.1) | 2,663 (21.9) | | | | 3,980 (44.8) | 4,012 (49.8) | 7,992 (47.2) |
| College/ university | 323 (5.0) | 501 (8.8) | 824 (6.8) | | | | 2,067 (23.3) | 2,502 (31.0) | 4,569 (27.0) |
| Marital status[a] | | | | | | | | | |
| No living spouse or partner | 4,024 (52.7) | 1,641 (25.5) | 5,665 (40.3) | 3,135 (49.1) | 1,327 (24.1) | 4,462 (37.5) | 3,933 (43.7) | 2,137 (26.3) | 6,070 (35.5) |
| Living spouse or partner | 3,609 (47.3) | 4,798 (74.5) | 8,407 (59.7) | 3,247 (50.9) | 4,185 (75.9) | 7,432 (62.5) | 5,058 (56.3) | 5,980 (73.7) | 11,038 (64.5) |
| Living in | | | | | | | | | |
| Urban area | 4,454 (58.3) | 3,755 (58.3) | 8,209 (58.3) | 3,873 (61.3) | 3,331 (61.0) | 7,204 (61.2) | 5,770 (64.1) | 5,164 (63.6) | 10,934 (63.9) |
| Rural area | 3,190 (41.7) | 2,691 (41.7) | 5,881 (41.7) | 2,445 (38.7) | 2,128 (39.0) | 4,573 (38.8) | 3,225 (35.9) | 2,957 (36.4) | 6,182 (36.1) |
| Smoking[a] | | | | | | | | | |
| Never smoked | 4,879 (67.0) | 1,374 (21.7) | 6,253 (45.9) | 3,175 (53.1) | 1,465 (27.5) | 4,640 (41.0) | 3,490 (39.3) | 2,626 (32.5) | 6,116 (36.1) |
| Previously smoked | 1,365 (18.7) | 3,420 (54.1) | 4,785 (35.2) | 1,846 (30.9) | 2,914 (54.7) | 4,760 (42.1) | 4,524 (50.9) | 4,862 (60.2) | 9,386 (55.3) |
| Smoker | 1,040 (14.3) | 1,531 (24.2) | 2,571 (18.9 | 959 (16.0) | 948 (17.8) | 1,907 (16.9) | 874 (9.8) | 588 (7.3) | 1,462 (8.6) |
| Income after taxes (NOK)[a,b] | | | | | | | | | |
| Mean | 78,324 | 114,525 | 94,932 | 164,192 | 227,741 | 193,635 | 265,526 | 355,139 | 308,033 |
| (SD) | (35,433) | (141,852) | (101,171) | (79,654) | (113,490) | (101,863) | (99,513) | (293,334) | (219,135) |
| Median | 72,169 | 102,226 | 84,392 | 150,678 | 206,732 | 176,371 | 247,091 | 308,704 | 276,159 |
| (Q₁; Q₃) | (56,803; 90,002) | (81,470; 130,230) | (66,523; 110,778) | (117,957; 188,817) | (170,777; 255,863) | (137,911; 226,011) | (208,279; 300,010) | (262,552; 375,730) | (229,227; 336,789) |

Abbreviations: HUNT = Trøndelag Health Study; NOK = Norwegian kroner; Q1; Q3 = first and third quartile

[a]Numbers do not add up to n = 7,644/6,446/14,090 (HUNT2), n = 6,388/5,515/11,903 (HUNT3), and n = 9,000/8,124/17,124 (HUNT4) due to missing information;

[b]Income after taxes, values of 0- or negative income for the year of participation were replaced by average of the remaining two values (or one value if only single value available), 0-income for all three years were replaced with missing.

prevalence of both frequent and risk drinking was generally low in all three HUNT surveys, but increased among both women and men from HUNT2 to HUNT4. The prevalence increased more from HUNT2 to HUNT3, than from HUNT3 to HUNT4.

## Prevalence and changes in alcohol consumption assessed by PEth

Table 5 shows detailed information about the prevalence of different PEth concentrations in HUNT3 and HUNT4. In HUNT4 the prevalence of PEth <0.03, >0.06, >0.10, and >0.30 µmol/l was 68.1%, 21.2%, 14.3%, and 4.3%, respectively. Fig 1f–1i illustrates graphically

**Table 4. Prevalence and change in prevalence of self-reported alcohol consumption among older adults (≥65 years) through three independent surveys (HUNT2, HUNT3, and HUNT4).** Numbers are frequencies (%) unless stated otherwise. *Italic font* represents numbers stratified by age groups.

| | HUNT2 1995–97 | HUNT3 2006–08 | HUNT4 2017–19 |
|---|---|---|---|
| **Lifetime abstaining**[a]* | | | |
| N | 13,617 | 11,196 | 16,676 |
| Total | 4,202 (30.9) | 983 (8.8) | 866 (5.2) |
| *65–69* | *894 (21.4)* | *206 (5.2)* | *143 (2.6)* |
| *70–74* | *1,143 (28.6)* | *253 (8.5)* | *207 (4.2)* |
| *≥75* | *2,165 (39.8)* | *524 (12.3)* | *516 (8.3)* |
| Women | 3,050 (41.6) | 754 (12.9) | 664 (7.6) |
| *65–69* | *645 (30.1)* | *153 (7.5)* | *96 (3.5)* |
| *70–74* | *823 (40.0)* | *198 (13.2)* | *155 (6.1)* |
| *≥75* | *1,582 (50.5)* | *403 (17.5)* | *413 (12.2)* |
| Men | 1,152 (18.3) | 229 (4.3) | 202 (2.5) |
| *65–69* | *249 (12.2)* | *53 (2.7)* | *47 (1.7)* |
| *70–74* | *320 (16.5)* | *55 (3.7)* | *52 (2.2)* |
| *≥75* | *583 (25.3)* | *121 (6.2)* | *103 (3.6)* |
| **Former drinking**[b]** | | | |
| N | | 11,196 | 16,676 |
| Total | No data | 1,044 (9.3) | 2,373 (14.2) |
| *65–69* | | *237 (6.0)* | *498 (9.1)* |
| *70–74* | | *235 (7.9)* | *563 (11.4)* |
| *≥75* | | *572 (13.5)* | *1,312 (21.1)* |
| Women | | 641 (11.0) | 1,526 (17.5) |
| *65–69* | | *145 (7.1)* | *319 (11.5)* |
| *70–74* | | *138 (9.2)* | *351 (13.8)* |
| *≥75* | | *358 (15.5)* | *856 (25.3)* |
| Men | | 403 (7.5) | 847 (10.6) |
| *65–69* | | *92 (4.8* | *179 (6.6)* |
| *70–74* | | *97 (6.6)* | *212 (8.8)* |
| *≥75* | | *214 (11.0)* | *456 (16.0)* |
| **Current drinking**[c]** | | | |
| N | 8,680 | 11,196 | 16,676 |
| Total | 3,696 (42.6) | 9,169 (81.9) | 13,437 (80.6) |
| *65–69* | *1,591 (53.3)* | *3,537 (88.9)* | *4,847 (88.3)* |
| *70–74* | *1,161 (44.5)* | *2,483 (83.6)* | *4,190 (84.5)* |
| *≥75* | *944 (30.6)* | *3,149 (74.2)* | *4,400 (70.6)* |
| Women | 1,152 (29.1) | 4,454 (76.1) | 6,518 (74.9) |
| *65–69* | *528 (39.7)* | *1,751 (85.5)* | *2,358 (85.0)* |
| *70–74* | *336 (29.4)* | *1,159 (77.5)* | *2,043 (80.1)* |
| *≥75* | *288 (19.4)* | *1,544 (67.0)* | *2,117 (62.5)* |
| Men | 2,544 (53.9) | 4,715 (88.2) | 6,919 (86.8) |
| *65–69* | *1,063 (64.3)* | *1,786 (92.5)* | *2,489 (91.7)* |
| *70–74* | *825 (56.2)* | *1,324 (89.7)* | *2,147 (89.1)* |
| *≥75* | *656 (41.0)* | *1,605 (82.7)* | *2,283 (80.3)* |
| **Frequent drinking**[d]** | | | |
| N | 8,680 | 11,196 | 16,676 |
| Total | 100 (1.2) | 365 (3.3) | 738 (4.4) |
| *65–69* | *26 (0.9)* | *158 (4.0)* | *231 (4.2)* |

*(Continued)*

**Table 4.** (Continued)

| | HUNT2 1995–97 | HUNT3 2006–08 | HUNT4 2017–19 |
|---|---|---|---|
| *70–74* | *30 (1.1)* | *100 (3.4)* | *274 (5.5)* |
| *≥75* | *44 (1.4)* | *107 (2.5)* | *233 (3.7)* |
| Women | 19 (0.5) | 126 (2.2) | 268 (3.1) |
| *65–69* | *4 (0.3)* | *58 (2.8)* | *85 (3.1)* |
| *70–74* | *4 (0.4)* | *34 (2.3)* | *93 (3.6)* |
| *≥75* | *11 (0.7)* | *34 (1.5)* | *90 (2.7)* |
| Men | 81 (1.7) | 239 (4.5) | 470 (5.9) |
| *65–69* | *22 (1.3)* | *100 (5.2)* | *146 (5.4)* |
| *70–74* | *26 (1.8)* | *66 (4.5)* | *181 (7.5)* |
| *≥75* | *33 (2.1)* | *73 (3.8)* | *143 (5.0)* |
| **Risk drinking**[e]** | | | |
| N | 13,491 | 10,125 | 14,408 |
| Total | 144 (1.1) | 321 (3.2) | 805 (5.6) |
| *65–69* | *57 (1.4)* | *151 (4.1)* | *362 (7.2)* |
| *70–74* | *45 (1.1)* | *91 (3.4)* | *264 (6.0)* |
| *≥75* | *42 (0.8)* | *79 (2.1)* | *179 (3.6)* |
| Women | 22 (0.3) | 70 (1.3) | 178 (2.5) |
| *65–69* | *8 (0.4)* | *36 (1.9)* | *82 (3.3)* |
| *70–74* | *7 (0.3)* | *20 (1.5)* | *54 (2.5)* |
| *≥75* | *7 (0.2)* | *14 (0.7)* | *42 (1.6)* |
| Men | 122 (2.0) | 251 (5.1) | 627 (8.7) |
| *65–69* | *49 (2.5)* | *115 (6.3)* | *280 (10.9)* |
| *70–74* | *38 (2.0)* | *71 (5.2)* | *210 (9.4)* |
| *≥75* | *35 (1.5)* | *65 (3.9)* | *137 (5.6)* |

Abbreviations: HUNT = Trøndelag Health Study; N = number

[a]Lifetime abstinence: HUNT2 = total abstinence from alcohol; HUNT3 and HUNT4 = never consumed alcohol;

[b]Former drinking: HUNT3 and HUNT4 = not consumed alcohol during the past year;

[c]Current drinking: Consumed alcohol ≥once per month or ≥few times per year;

[d]Frequent drinking: Drinking alcohol ≥4 times per week;

[e]Risk drinking: Drinking ≥8 units of alcohol (≥96 g alcohol) per week.

*Not adjusted for cluster effect due to repeated measurements;

**Adjusted for cluster effect due to repeated measurement

the crude odds for different patterns of alcohol consumption assessed by PEth in HUNT3 and HUNT4 stratified by gender and age.

The prevalence of PEth <0.03 μmol/l decreased from HUNT3 and HUNT4, and decreased more among women than men. The prevalence of PEth >0.06 μmol/l increased from HUNT3 to HUNT4 in both genders, while the prevalence of PEth >0.30 μmol/l decreased slightly from HUNT3 to HUNT4.

## Gender differences within and between the HUNT surveys

Tables 6 and 7 show the gender differences in the prevalence of self-reported alcohol consumption and alcohol consumption assessed by PEth within the HUNT surveys, both tables stratified by age. Men in all three age groups were less likely to be lifetime abstainers and more likely to be current drinkers, frequent drinkers, and risk drinkers than women in all three

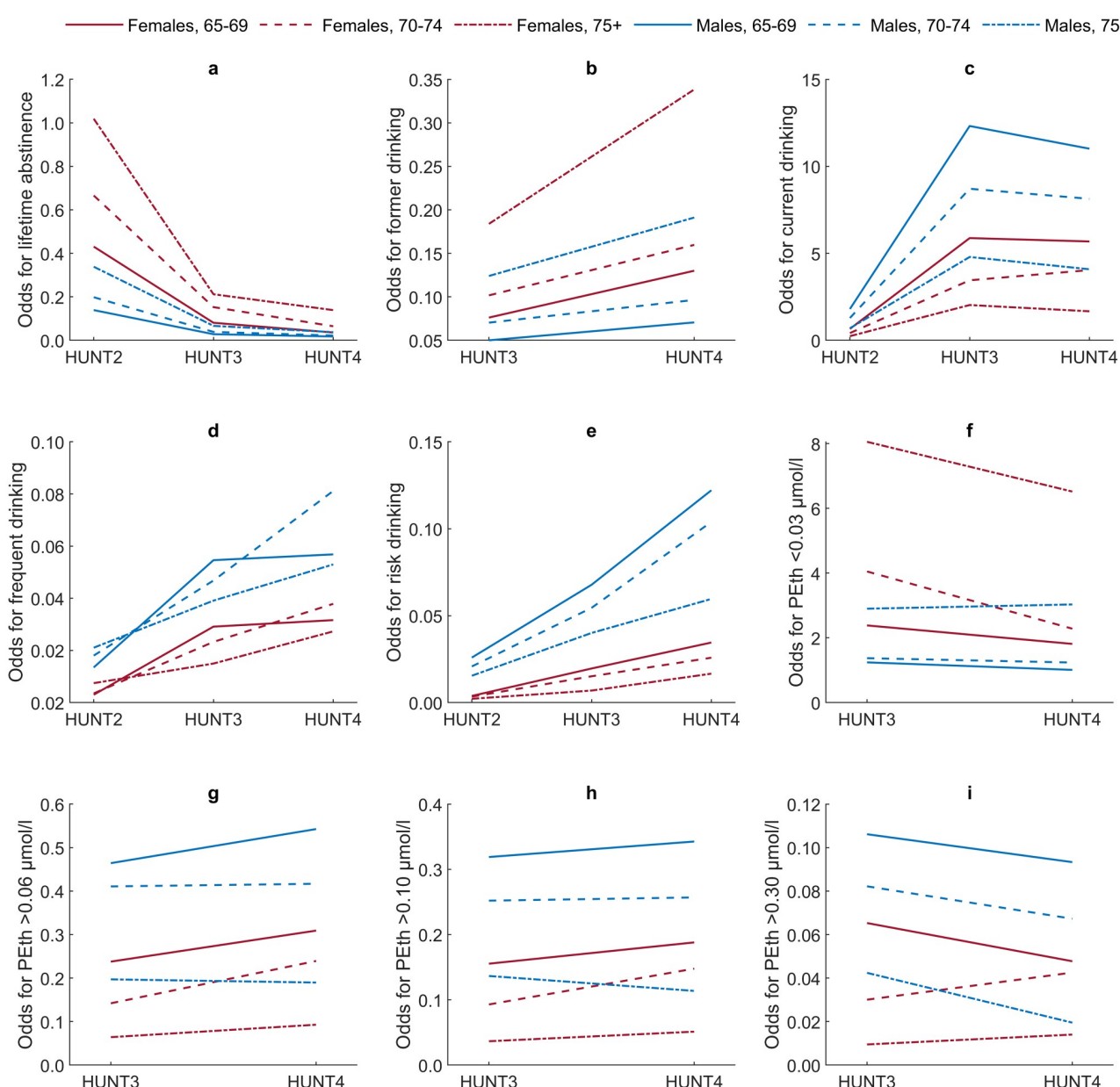

**Fig 1. a-i**. Crude odds for different patterns of alcohol consumption assessed by self-report (HUNT2, HUNT3, and HUNT4) and Phosphatidylethanol 16:0/18:1 (PEth) (HUNT3 and HUNT4). Crude odds for 1a) lifetime abstinence; 1b) former drinking; 1c) current drinking; 1d) frequent drinking; 1e) risk drinking; 1f) PEth <0.03 μmol/l; 1g) PEth >0.06 μmol/l; 1h) PEth >0.10 μmol/l; 1i) PEth >0.30 μmol/l, among older women and men in different age groups (65–69, 70–74, and ≥75 years).

HUNT surveys. In HUNT3 and HUNT4, men were less likely than women to have PEth <0.03 μmol/l and more likely to have PEth >0.06 μmol/l and >0.10 μmol/l in all three age groups. Men compared to women had more often PEth >0.30 μmol/l in both HUNT3 and HUNT4, for all age groups, but not for those aged 75 years and older in HUNT4.

Tables 6 and 7 show the gender differences in the prevalence of self-reported alcohol consumption and alcohol consumption assessed by PEth between the HUNT surveys, both tables stratified by age. The gender differences for abstinence did not change between the three

**Table 5. Prevalence and change in prevalence in pattern of alcohol consumption measured by Phosphatidyletha-nol 16:0/18:1 (PEth) in older people (≥65 years) through two independent surveys (HUNT3 and HUNT4).** Numbers are frequencies (%) unless stated otherwise. *Italic font* represents numbers stratified by age groups.

| | HUNT3 (2006–08 | HUNT4 (2017–19 |
|---|---|---|
| **PEth <0.03 μmol/l[a] *** | | |
| N | 6,068 | 7,290 |
| Total | 4,370 (72.0) | 4,967 (68.1) |
| *65–69* | *1,359 (63.0)* | *1,361 (57.5)* |
| *70–74* | *1,089 (69.0)* | *1,346 (62.6)* |
| *≥75* | *1,922 (82.3)* | *2,260 (81.5)* |
| Women | 2,538 (80.4) | 2,862 (74.7) |
| *65–69* | *770 (70.4)* | *778 (64.5)* |
| *70–74* | *632 (80.2)* | *767 (69.5)* |
| *≥75* | *1,136 (89.0)* | *1,317 (86.7)* |
| Men | 1,832 (63.0) | 2,105 (60.8) |
| *65–69* | *589 (55.4)* | *583 (50.3)* |
| *70–74* | *457 (57.8)* | *579 (55.3)* |
| *≥75* | *786 (74.4)* | *943 (75.2)* |
| **PEth >0.06 μmol/l[b] *** | | |
| N | 6,068 | 7,290 |
| Total | 1,126 (18.6) | 1,543 (21.2) |
| *65–69* | *547 (25.4)* | *693 (29.3)* |
| *70–74* | *328 (20.8)* | *521 (24.2)* |
| *≥75* | *251 (10.8)* | *329 (11.9)* |
| Women | 385 (12.2) | 627 (16.4) |
| *65–69* | *210 (19.2)* | *285 (23.6)* |
| *70–74* | *98 (12.4)* | *213 (19.3)* |
| *≥75* | *77 (6.0)* | *129 (8.5)* |
| Men | 741 (25.5) | 916 (26.5) |
| *65–69* | *337 (31.7)* | *408 (35.2)* |
| *70–74* | *230 (29.1)* | *308 (29.4)* |
| *≥75* | *174 (16.5)* | *200 (15.9)* |
| **PEth >0.10 μmol/l[c] *** | | |
| N | 6,068 | 7,290 |
| Total | 802 (13.2) | 1,045 (14.3) |
| *65–69* | *404 (18.7)* | *487 (20.6)* |
| *70–74* | *226 (14.3)* | *356 (16.6)* |
| *≥75* | *172 (7.4)* | *202 (7.3)* |
| Women | 259 (8.2) | 407 (10.6) |
| *65–69* | *147 (13.4)* | *191 (15.8)* |
| *70–74* | *67 (8.5)* | *142 (12.9)* |
| *≥75* | *45 (3.5)* | *74 (4.9)* |
| Men | 543 (18.7) | 638 (18.4) |
| *65–69* | *257 (24.2)* | *296 (25.5)* |
| *70–74* | *159 (20.1)* | *214 (20.4)* |
| *≥75* | *127 (12.0)* | *128 (10.2)* |
| **PEth >0.30 μmol/l[d] *** | | |
| N | 6,068 | 7,290 |
| Total | 307 (5.1) | 310 (4.3) |
| *65–69* | *169 (7.8)* | *154 (6.5)* |

(*Continued*)

**Table 5.** (Continued)

|  | HUNT3 (2006–08 | HUNT4 (2017–19) |
|---|---|---|
| *70–74* | *83 (5.3)* | *111 (5.2)* |
| *≥75* | *55 (2.4)* | *45 (1.6)* |
| Women | 102 (3.2) | 121 (3.2) |
| *65–69* | *67 (6.1)* | *55 (4.6)* |
| *70–74* | *23 (2.9)* | *45 (4.1)* |
| *≥75* | *12 (0.9)* | *21 (1.4)* |
| Men | 205 (7.0) | 189 (5.5) |
| *65–69* | *102 (9.6)* | *99 (8.5)* |
| *70–74* | *60 (7.6)* | *66 (6.3)* |
| *≥75* | *43 (4.1)* | *24 (1.9)* |

Abbreviations: HUNT = Trøndelag Health Study; N = number; PEth = Phosphatidylethanol 16:0/18:1

[a]PEth <0.03 μmol/l = abstinence or a very low level of alcohol consumption;

[b]PEth >0.06 μmol/l = consumption of more than one alcohol unit/day;

[c]PEth >0.10 μmol/l = consumption of more than three alcohol units/day;

[d]PEth >0.30 μmol/l = heavy alcohol consumption.

*Not adjusted for cluster effect due to repeated measurements.

HUNT surveys, except for the age group 65–69 years, where the gender differences were reduced between HUNT2 and HUNT4. For current drinkers, gender differences were higher in HUNT2 than in HUNT3 and HUNT4 in all age categories. There were no gender differences for frequent drinkers and risk drinkers between surveys. For all PEth concentrations (<0.03, >0.06, >0.10, and >0.30 μmol/l) the gender differences were reduced from HUNT3 to HUNT4 either among those aged 70–74 years and/or those aged 75 years and older.

## Discussion

In this large Norwegian population-based health study, we examined the prevalence and changes in alcohol consumption among older adults (≥65 years) by gender and age using self-report measures in three independent HUNT surveys (HUNT2 1995–97, HUNT3 2006–08, and HUNT4 2017–19), and using an objective measure of alcohol consumption (PEth) in two of the same surveys (HUNT3 2006–08 and HUNT4 2017–19). The self-reported prevalence of abstinence decreased over time while the prevalence of self-reported frequent drinking and risk drinking increased for both genders over time. Also, the prevalence of PEth <0.03 μmol/l decreased, and the prevalence of PEth >0.06 μmol/l increased from HUNT3 to HUNT4 in women and men. However, the prevalence of PEth >0.30 μmol/l (indicating very high consumption) was slightly lower in HUNT4 compared to HUNT3. Men were less likely to be abstaining from alcohol and more likely to be current drinkers and risk drinkers compared to women, independent of health survey, age group, and the method used to assess alcohol consumption. The gender differences were reduced over time, and this were found more often when gender differences were assessed with PEth than self-reported.

A main finding in the present study was the decrease in the prevalence of self-reported lifetime abstention and the increase in the prevalence of frequent and risk drinking during the study period from 1995 to 2019. We also found that the prevalence of PEth <0.03 μmol/l decreased and PEth >0.06 μmol/l increased from HUNT3 (2006–08) to HUNT4 (2017–19). These changes in older adults' drinking patterns are in accordance with previous studies from

**Table 6. Gender differences[a] and change in gender differences over time in pattern of self-reported alcohol consumption among older adults (≥65 years) through three independent surveys (HUNT2, HUNT3, and HUNT4).**

| | Gender differences within HUNT2, men vs. women, OR (95% CI) | Gender differences within HUNT3, men vs. women, OR (95% CI) | Gender differences within HUNT4, men vs. women, OR (95% CI) |
|---|---|---|---|
| **Lifetime abstaining[b]*** | | | |
| Crude | | | |
| 65–69 | 0.32 (0.27; 0.38) | 0.35 (0.25; 0.48) | 0.49 (0.35; 0.70) |
| 70–74 | 0.30 (0.26; 0.35) | 0.25 (0.19; 0.35) | 0.34 (0.25; 0.47) |
| ≥75 | 0.33 (0.30; 0.37) | 0.31 (0.23; 0.42) | 0.27 (0.22; 0.34) |
| Adjusted | | | |
| 65–69 | 0.38 (0.32; 0.44) | 0.44 (0.32; 0.60) | 0.63 (0.44; 0.90) |
| 70–74 | 0.34 (0.29; 0.40) | 0.31 (0.23; 0.43) | 0.42 (0.30; 0.58) |
| ≥75 | 0.37 (0.33; 0.42) | 0.37 (0.30; 0.46) | 0.33 (0.26; 0.42) |
| **Former drinking[c]**** | | | |
| Crude | | | |
| 65–69 | No data | 0.66 (0.50; 0.86) | 0.54 (0.45; 0.66) |
| 70–74 | | 0.69 (0.53; 0.91) | 0.60 (0.50; 0.72) |
| ≥75 | | 0.67 (0.56; 0.81) | 0.56 (0.50; 0.64) |
| Adjusted | | | |
| 65–69 | | 0.89 (0.68; 1.16) | 0.76 (0.63; 0.93) |
| 70–74 | | 0.96 (0.73; 1.26) | 0.80 (0.67; 0.97) |
| ≥75 | | 0.94 (0.78; 1.13) | 0.79 (0.69; 0.90) |
| **Current drinking[d]**** | | | |
| Crude | | | |
| 65–69 | 2.74 (2.36; 3.18) | 2.10 (1.70; 2.58) | 1.94 (1.63; 2.30) |
| 70–74 | 3.07 (2.61; 3.62) | 2.53 (2.05; 3.11) | 2.01 (1.72; 2.37) |
| ≥75 | 2.89 (2.46; 3.40) | 2.36 (2.04; 2.73) | 2.45 (2.18; 2.75) |
| Adjusted | | | |
| 65–69 | 2.18 (1.87; 2.53) | 1.55 (1.25; 1.92) | 1.37 (1.15; 1.64) |
| 70–74 | 2.46 (2.08; 2.91) | 1.84 (1.49; 2.27) | 1.51 (1.28; 1.78) |
| ≥75 | 2.32 (1.96; 2.73) | 1.75 (1.50; 2.03) | 1.79 (1.59; 2.02) |
| **Frequent drinking[e]**** | | | |
| Crude | | | |
| 65–69 | 4.47 (1.54; 13.01) | 1.87 (1.35; 2.61) | 1.80 (1.37; 2.36) |
| 70–74 | 5.13 (1.78; 14.73) | 2.01 (1.32; 3.06) | 2.14 (1.66; 2.77) |
| ≥75 | 2.82 (1.42; 5.61) | 2.61 (1.73; 3.94) | 1.94 (1.48; 2.54) |
| Adjusted | | | |
| 65–69 | 4.11 (1.41; 11.96) | 1.74 (1.25; 2.42) | 1.61 (1.22; 2.12) |
| 70–74 | 4.51 (1.56; 12.99) | 1.81 (1.19; 2.76) | 1.97 (1.52; 2.56) |
| ≥75 | 2.47 (1.27; 4.91) | 2.25 (1.49; 3.41) | 1.71 (1.30; 2.24) |
| **Risk drinking[f]**** | | | |
| Crude | | | |
| 65–69 | 6.78 (3.20; 14.35) | 3.46 (2.36; 5.05) | 3.54 (2.75; 4.55) |
| 70–74 | 6.11 (2.72; 13.71) | 3.60 (2.18; 5.95) | 4.03 (2.97; 5.46) |
| ≥75 | 7.07 (3.14; 15.95) | 5.79 (3.24; 10.35) | 3.58 (2.52; 5.09) |
| Adjusted | | | |
| 65–69 | 6.49 (3.06; 13.74) | 3.30 (2.25; 4.83) | 3.28 (2.54; 4.23) |
| 70–74 | 5.78 (2.57; 13.01) | 3.44 (2.08; 5.69) | 3.83 (2.82; 5.21) |
| ≥75 | 6.88 (3.04; 15.54) | 5.55 (3.10; 9.95) | 3.44 (2.41; 4.89) |
| **Differences between surveys in gender (men vs. women) differences** | | | |

*(Continued)*

**Table 6.** (Continued)

| | Gender differences within HUNT2, men vs. women, OR (95% CI) | Gender differences within HUNT3, men vs. women, OR (95% CI) | Gender differences within HUNT4, men vs. women, OR (95% CI) |
|---|---|---|---|
| | HUNT2 vs. HUNT3 | HUNT2 vs. HUNT4 | HUNT3 vs. HUNT4 |
| **Lifetime abstaining[b]*** | | | |
| Crude | | | |
| 65–69 | 0.92 (0.65; 1.32) | **0.66 (0.45; 0.97)** | 0.71 (0.44; 1.15) |
| 70–74 | 1.17 (0.83; 1.65) | 0.87 (0.61; 1.24) | 0.74 (0.48; 1.16) |
| ≥75 | 1.06 (0.83; 1.35) | 1.23 (0.95; 1.58) | 1.16 (0.85; 1.58) |
| Adjusted | | | |
| 65–69 | 0.87 (0.60; 1.24) | **0.60 (0.40; 0.89)** | 0.69 (0.43; 1.12) |
| 70–74 | 1.09 (0.77; 1.54) | 0.81 (0.57; 1.16) | 0.75 (0.48; 1.17) |
| ≥75 | 1.00 (0.78; 1.28) | 1.13 (0.88; 1.45) | 1.13 (0.83; 1.54) |
| **Former drinking[c]**** | | | |
| Crude | | | |
| 65–69 | No data | No data | 1.21 (0.87; 1.68) |
| 70–74 | | | 1.15 (0.83; 1.59) |
| ≥75 | | | 1.19 (0.96; 1.49) |
| Adjusted | | | |
| 65–69 | | | 1.16 (0.83; 1.62) |
| 70–74 | | | 1.19 (0.86; 1.65) |
| ≥75 | | | 1.19 (0.95; 1.49) |
| **Current drinking[d]**** | | | |
| Crude | | | |
| 65–69 | **1.31 (1.01; 1.69)** | **1.41 (1.12; 1.77)** | 1.08 (0.83; 1.42) |
| 70–74 | 1.22 (0.93; 1.59) | **1.53 (1.21; 1.92)** | 1.25 (0.96; 1.63) |
| ≥75 | 1.22 (0.98; 1.52) | 1.18 (0.97; 1.44) | 0.96 (0.80; 1.16) |
| Adjusted | | | |
| 65–69 | **1.40 (1.08; 1.82)** | **1.58 (1.26; 2.00)** | 1.13 (0.86; 1.48) |
| 70–74 | **1.33 (1.02; 1.74)** | **1.63 (1.29; 2.05)** | 1.22 (0.94; 1.59) |
| ≥75 | **1.33 (1.06; 1.65)** | **1.29 (1.06; 1.58)** | 0.97 (0.81; 1.17) |
| **Frequent drinking[e]**** | | | |
| Crude | | | |
| 65–69 | 2.39 (0.78; 7.29) | 2.49 (0.83; 7.49) | 1.04 (0.68; 1.60) |
| 70–74 | 2.55 (0.82; 7.94) | 2.39 (0.81; 7.09) | 0.94 (0.57; 1.54) |
| ≥75 | 1.08 (0.49; 2.41) | 1.46 (0.70; 3.04) | 1.35 (0.82; 2.20) |
| Adjusted | | | |
| 65–69 | 2.36 (0.77; 7.22) | 2.55 (0.85; 7.69) | 1.08 (0.70; 1.66) |
| 70–74 | 2.49 (0.80; 7.78) | 2.28 (0.77; 6.78) | 0.92 (0.56; 1.50) |
| ≥75 | 1.10 (0.49; 2.44) | 1.45 (0.69; 3.02) | 1.32 (0.81; 2.16) |
| **Risk drinking[f]**** | | | |
| Crude | | | |
| 65–69 | 1.96 (0.85; 4.55) | 1.92 (0.87; 4.23) | 0.98 (0.62; 1.54) |
| 70–74 | 1.70 (0.65; 4.39) | 1.52 (0.64; 3.60) | 0.89 (0.50; 1.61) |
| ≥75 | 1.22 (0.45; 3.32) | 1.97 (0.81; 4.78) | 1.61 (0.82; 3.18) |
| Adjusted | | | |
| 65–69 | 1.97 (0.85; 4.56) | 1.98 (0.90; 4.37) | 1.01 (0.64; 1.59) |
| 70–74 | 1.68 (0.65; 4.36) | 1.51 (0.63; 3.58) | 0.90 (0.50; 1.61) |

*(Continued)*

**Table 6.** (Continued)

| | Gender differences within HUNT2, men vs. women, OR (95% CI) | Gender differences within HUNT3, men vs. women, OR (95% CI) | Gender differences within HUNT4, men vs. women, OR (95% CI) |
|---|---|---|---|
| ≥75 | 1.24 (0.46; 3.37) | 2.00 (0.83; 4.85) | 1.62 (0.82; 3.18) |

Abbreviations: CI = Confidence interval; HUNT = Trøndelag Health Study; N = number; OR = Odds Ratio

[a]Gender differences reported as OR with 95% CI, crude and adjusted for income after taxes and marital status;

[b]Lifetime abstinence: HUNT2 = total abstinence from alcohol; HUNT3 and HUNT4 = never consumed alcohol;

[c]Former drinking: HUNT3 and HUNT4 = not consumed alcohol during the past year;

[d]Current drinking: Consumed alcohol ≥once per month or ≥few times per year;

[e]Frequent drinking: Drinking alcohol ≥4 times per week;

[f]Risk drinking: Drinking ≥8 units of alcohol (≥96 g alcohol) per week.

*Not adjusted for cluster effect due to repeated measurements;

**Adjusted for cluster effect due to repeated measurement

Norway [7, 9], Sweden [12, 69], and the USA [70], but in contrast to those of older adults in Denmark [69], Finland [19], Germany [16], Spain [18], and Australia [20], where drinking frequency and risk drinking have been quite stable or decreased in recent decades. Even so, older Norwegian seem to drink less frequently and with less risk than European older adults. In HUNT4, only 4.4% and 5.6% of the participants self-reported that they drank frequently (≥4 times/week) and had a risky drinking pattern (≥8 units of alcohol per week), which is lower than the prevalence of frequent drinking (16.4–39.0%) [6, 10] and risk drinking (21.0–46.5%) [8, 10, 11] found among older adults in several European countries (Finland, Denmark, Netherland, Belgium, and Portugal). Differences in drinking frequency and risk drinking between countries may be due to social norms and drinking cultures [10, 11].

In our study, the increase in self-reported alcohol consumption was particularly large between 1995–97 (HUNT2) and 2006–08 (HUNT3), and levelled out more between 2006–08 (HUNT3) and 2017–19 (HUNT4), and a similar finding was confirmed by another Norwegian study using data from the general older population [7]. We also found a slight decrease in the prevalence of PEth >0.30 μmol/l (indicating heavy consumption) from HUNT3 to HUNT4.

In recent decades, several contextual changes have occurred that could partly explain the increase in alcohol consumption among older adults [15]. In Norway, alcoholic beverages have become more readily available as a result of the increasing number of places selling alcohol, increased opening hours in pubs and restaurants serving alcohol, and easier access to neighboring countries with lower alcohol prices [17, 71]. Higher education and increased income (which has made alcoholic beverages more affordable), and international traveling with adaption to a more continental drinking pattern and extended international Tax Free shopping, might also explain the increased alcohol consumption among Norwegian older adults [17, 32]. Further, better health, stronger focus on pleasure and self-realization, more positive attitudes to alcohol consumption, and more leisure time than for previous generations of older adults may have contributed to the change in today's generation of older adults [7, 9, 72, 73].

In our study, most of the increase in self-reported alcohol consumption appeared for both women and men. However, gender differences in self-reported abstinence and current drinking decreased through the three HUNT surveys for some or all age groups. This is in accordance with previous studies conducted in Norway [7, 9] and other Western countries [10, 12, 13]. The findings that self-reported frequent drinking (≥4 times per week) and risk drinking (≥8 units of alcohol per week) increased at the same level among women and men throughout

**Table 7. Gender differences[a] and change in gender differences over time in pattern of alcohol consumption measured by Phosphatidylethanol 16:0/18:1 (PEth) in older people (≥65 years) through two independent surveys (HUNT3 and HUNT4).** Numbers are frequencies (%) unless stated otherwise. *Italic font* represents numbers stratified by age groups.

| | Gender differences within HUNT3, men vs. women, OR (95% CI) | Gender differences within HUNT4, men vs. women, OR (95% CI) | Differences between surveys in gender differences, men vs. women, HUNT3 vs HUNT4, OR (95% CI) |
|---|---|---|---|
| **PEth<0.03 μmol/l[b]\*** | | | |
| Crude | | | |
| 65–69 | **0.52 (0.44; 0.62)** | **0.56 (0.47; 0.66)** | 0.94 (0.73; 1.19) |
| 70–74 | **0.34 (0.27; 0.42)** | **0.54 (0.45; 0.65)** | **0.63 (0.47; 0.83)** |
| ≥75 | **0.36 (0.29; 0.45)** | **0.47 (0.38; 0.57)** | 0.77 (0.58; 1.04) |
| Adjusted | | | |
| 65–69 | **0.58 (0.48; 0.69)** | **0.64 (0.54; 0.76)** | 0.90 (0.70; 1.15) |
| 70–74 | **0.38 (0.30; 0.48)** | **0.60 (0.50; 0.72)** | **0.63 (0.47; 0.84)** |
| ≥75 | **0.40 (0.32; 0.50)** | **0.52 (0.42; 0.63)** | 0.77 (0.58; 1.04) |
| **PEth>0.06 μmol/l[c]\*** | | | |
| Crude | | | |
| 65–69 | **1.95 (1.60; 2.38)** | **1.76 (1.47; 2.10)** | 1.11 (0.85; 1.45) |
| 70–74 | **2.89 (2.23; 3.76)** | **1.74 (1.43; 2.13)** | **1.66 (1.19; 2.31)** |
| ≥75 | **3.07 (2.32; 4.07)** | **2.04 (1.62; 2.59)** | **1.50 (1.04; 2.17)** |
| Adjusted | | | |
| 65–69 | **1.85 (1.51; 2.26)** | **1.61 (1.34; 1.93)** | 1.15 (0.88; 1.50) |
| 70–74 | **2.73 (2.10; 3.55)** | **1.63 (1.33; 2.00)** | **1.67 (1.20; 2.32)** |
| ≥75 | **2.95 (2.22; 3.92)** | **1.96 (1.54; 2.48)** | **1.51 (1.04; 2.18)** |
| **PEth>0.10 μmol/l[d]\*** | | | |
| Crude | | | |
| 65–69 | **2.05 (1.64; 2.57)** | **1.82 (1.49; 2.23)** | 1.13 (0.83; 1.52) |
| 70–74 | **2.71 (2.00; 3.68)** | **1.74 (1.38; 2.19)** | **1.56 (1.06; 2.29)** |
| ≥75 | **3.74 (2.63; 5.31)** | **2.22 (1.65; 2.99)** | **1.68 (1.06; 2.67)** |
| Adjusted | | | |
| 65–69 | **1.97 (1.57; 2.47)** | **1.71 (1.39; 2.10)** | 1.15 (0.85; 1.56) |
| 70–74 | **2.60 (1.92; 3.54)** | **1.63 (1.29; 2.06)** | **1.60 (1.09; 2.34)** |
| ≥75 | **3.65 (2.56; 5.20)** | **2.16 (1.60; 2.91)** | **1.69 (1.07; 2.68)** |
| **PEth>0.30 μmol/l[e]\*** | | | |
| Crude | | | |
| 65–69 | **1.63 (1.18; 2.24)** | **1.95 (1.39; 2.75)** | 0.83 (0.52; 1.33) |
| 70–74 | **2.73 (1.67; 4.47)** | **1.58 (1.07; 2.33)** | 1.73 (0.92; 3.32) |
| ≥75 | **4.47 (2.34; 8.52)** | 1.39 (0.77; 2.51) | **3.21 (1.34; 7.70)** |
| Adjusted | | | |
| 65–69 | **1.63 (1.18; 2.24)** | **1.90 (1.35; 2.68)** | 0.86 (0.54; 1.37) |
| 70–74 | **2.80 (1.71; 4.59)** | **1.52 (1.03; 2.25)** | 1.84 (0.98; 3.45) |
| ≥75 | **4.78 (2.50; 9.12)** | 1.46 (0.81; 2.64) | **3.26 (1.36; 7.83)** |

Abbreviations: CI = Confidence interval; HUNT = Trøndelag Health Study; N = number; OR = Odds Ratio; PEth = Phosphatidylethanol 16:0/18:1

[a]Gender differences reported as OR with 95% CI, crude and adjusted for age, income after taxes, and marital status with women as reference category;

[b]PEth <0.03 μmol/l = abstinence or a very low level of alcohol consumption;

[c]PEth >0.06 μmol/l = consumption of more than one alcohol unit/day;

[d]PEth >0.10 μmol/l = consumption of more than three alcohol units/day;

[e]PEth >0.30 μmol/l = heavy alcohol consumption.

\*Not adjusted for cluster effect due to repeated measurements

the study period are not in line with reported findings from other Norwegian [7, 9] and Nordic studies [12, 69]. These studies reported a reduction in the gender gap in frequent drinking (≥2 times per week) [7, 9, 69] and risk drinking (≥100 g alcohol per week) [12] among older adults. However, different definitions of frequent and risk drinking complicate the comparison of gender convergence between studies. The use of different methods and definitions to measure alcohol consumption in our study might also explain that the gender differences in self-reported and objectively measured (PEth) alcohol consumption differed. Although we found no gender convergence in self-reported frequent and risk drinking, we did find gender convergence in the PEth concentrations considered to be high in the present study (>0.06, >0.10, and >0.30 μmol/l), and especially in the two oldest age groups (70+). The discrepancies in these findings may also be due to the fact that women are more likely to underreport alcohol consumption than men due to shame and guilt [74]. The gender convergence in self-reported current alcohol consumption and in objectively measured alcohol consumption may be due to a higher degree of gender equality for women who grew up after the Second World War, including improvement in the socioeconomic status, women's greater representation in the workplace, and a liberalization in attitudes towards women's drinking [7, 67, 75].

Although our data show a gender convergence in alcohol consumption, we found that men still drink more frequently and risky than women, assessed with both self-report and PEth (>0.06 and >0.10 μmol/l). This finding is also in line with other studies conducted in Norway [7, 9] and in Europe (Sweden, Denmark, Finland, the Netherlands, and Belgium) [10–12, 69, 76], and may be explained by women's greater vulnerability to alcohol [77], cultural values, traditional family structure, and gender roles where women are expected to drink moderately [32, 77].

In our study we also found that a higher proportion of older adults may have a risky drinking pattern according to the PEth concentration used in the present study. In total, 21.2% had a PEth concentration above 0.06 μmol/l in HUNT4, which indicates consuming one unit or more per day [56]. Even so, comparing self-reported alcohol consumption with PEth is complicated, as several factors affect the interpretation of this association [64, 78]. Firstly, self-reported alcohol consumption, and especially among heavy drinkers, may be underestimated in epidemiologic studies [42, 43], and PEth may to a larger extent identify older adults with a risky drinking pattern by detecting a more significant alcohol intake [79, 80]. Further, while the self-reported questions in HUNT3 and HUNT4 ask about the number of glasses of alcohol *usually* consumed in two weeks, PEth measures the *actual* intake of ethanol during the past 2–4 weeks [56]. We have no reason to believe that the study participants changed their alcohol consumption prior to inclusion in the study, but, due to random conditions, some may have had a deviant consumption, and this would complicate this comparison [56]. Also, a comparison of the results in our study is made difficult as PEth was measured in a subsample in HUNT3 and HUNT4 and not in the total sample, as was the case for the self-reported alcohol consumption.

## Clinical and public health implications

The increasing number of older adults with elevated alcohol consumption may become a public health challenge in the years to come [7]. Elevated alcohol consumption is a modifiable risk factor for several health conditions and a reduction in alcohol consumption may have great potential to reduce disease burden [21–23, 37]. Health care professionals could routinely ask about and assess older adults' alcohol consumption, and particularly when meeting older patients with insomnia and depression, falls and injuries, cognitive decline, hypertension, atrial fibrillation, liver disease, malnutrition, and frailty [4, 81, 82]. To reduce risk of alcohol

and drug interaction, health care professionals could also ask about older adults' alcohol consumption when administering or prescribing medications, and especially drugs with addiction potential [3, 31]. Older adults must be informed about their increased sensitivity and lower tolerance of alcohol and the possible risk associated with concurrent use of alcohol and medication [3, 31]. The gender convergence in alcohol consumption among older adults with women moving towards men's drinking patterns, highlights the need for health care professionals to inform older women about their increased risk of adverse effects of alcohol use [9, 41].

In older adults, the benefits of screening and brief intervention have been well documented [83–85], especially with regard to reduced alcohol consumption among hazardous and harmful drinkers [85]. There are several validated self-report questionnaires or screening tools available for the assessment of alcohol consumption [4, 57, 86, 87]. They are easy, non-invasive, and inexpensive, and can be applied in most health care settings without specialized health care professionals [78]. However, PEth can provide important additional information when alcohol consumption is denied or underestimated due to stigma, fear of sanctions, cognitive impairment, social desirability, or patients being uncooperative or unable to answer the questionnaire [88, 89]. Furthermore, ethical considerations may be raised when analyzing PEth in blood without consent in clinical settings. The main rule is therefore to receive informed consent from the person prior to blood sampling [90].

## Strengths and limitations

The main strength of the present study was the use of data from the past three HUNT surveys with repeated measurements dating back to the mid-1990s [48]. These three cross-sectional HUNT surveys were conducted in a stable homogeneous population located in the same geographical area (Nord-Trøndelag) with little migration, and strengthen the findings that changes in the prevalence of alcohol consumption are cultural and not the result of different people moving in or out of the area [17].

Another strength of our study was the use of PEth to objectively measure alcohol consumption, even if only in a subsample of two of the HUNT surveys. The use of PEth reduced the potential for recall bias and social desirability bias associated with self-reported alcohol consumption [80]. Few studies have examined PEth among older adults [44], and we need further studies that validate and examine the prevalence of different threshold values of PEth in an older population, examine the association between PEth and self-reported alcohol consumption, and examine the health consequences of different threshold values [56, 80]. The possible influence of body mass index and nutrition on the PEth results needs to be further explored [91, 92]. The present study did not have information about these factors.

Although the population of Nord-Trøndelag is assumed to be quite representative of the general population of Norway, Nord-Trøndelag does not have any large cities and has a lower proportion of immigrants, and the inhabitants have fewer years of education and lower mean incomes compared to the rest of Norway as a whole [48, 49, 51]. Thus, our findings may not be representative of the general population of older adults in Norway, and they may not correspond to the changes in alcohol consumption found among older adults in the most urban parts of Norway [9]. However, the general adult population of Trøndelag county seems to consume alcohol at the same level as the general adult population of Norway [71].

The participation rate has decreased from HUNT2 (69.5%) to HUNT3 (54.1%) and HUNT4 (54.0%) [48–50], and the non-participation rate in HUNT3 and HUNT4 (but not in HUNT2) was highest among the oldest age groups, among men, and among those with chronic diseases, poor self-rated health, and substance abuse problems [48, 50, 52, 93]. Thus, it

is likely that a lower proportion of older adults who were abstainers due to poor health [48, 50, 52] and of heavy drinkers [93] participated in HUNT3 and HUNT4, compared to HUNT2.

Further methodological limitations need attention. As indicated previously, not all drinking patterns (i.e., former drinking) or PEth were assessed in all three HUNT surveys. The questions about the number of glasses of wine, beer, or liquor consumed during the past two weeks have been kept unchanged across the surveys, which enables more valid results about risk drinking [48], whilst other questions have changed (i.e., the drinking frequency questionnaire). Furthermore, glasses of alcohol were converted to units of alcohol, and glasses of alcohol reported by the participants may not be equal to the volume of one standard unit of alcohol in Norway. For example, older adults may drink one half-liter glass of beer (500 ml) which contains about 1.5 units of alcohol (one unit of beer = 330 ml), or one glass of wine (200 ml) which contains about 1.6 units of alcohol (one unit of wine = 125 ml) [61]. Thus, the prevalence of risk drinking may be underestimated.

Also, as already mentioned, self-reported alcohol consumption is susceptible to underreporting, leading to the misclassification of some participants, which is known from previous studies [42, 43]. Underreporting and imprecise self-reporting of alcohol consumption among older adults may be due to memory errors [82], the stigma associated with drinking [28], and answering according to expectations of social desirability [94] and cultural norms [95]. Underestimation of alcohol consumption among older adults may also explain the discrepancies we found between the prevalence of self-reported alcohol consumption and of objectively measured alcohol consumption with PEth.

We lack a gold standard for alcohol consumption, both among the general [96] and older population [4], and the definition of risk drinking used in the present study [62] is not validated among older adults [4]. Future research should longitudinally validate and examine the association between different definitions of risk drinking and health consequences in older women and men [4].

## Conclusions

Among older adults ($\geq$65 years) in Norway, the self-reported prevalence of abstinence decreased while the self-reported prevalence of frequent drinking and risk drinking increased for both genders over a 24-year period (HUNT2-HUNT4). Also, the prevalence of PEth <0.03 μmol/l decreased, and the prevalence of PEth >0.06 μmol/l increased from HUNT3 (2006–08) to HUNT4 (2017–19) in women and men. However, the prevalence of PEth >0.30 μmol/l (indicating very high consumption) was lower in HUNT4 compared to HUNT3. Men were less likely to be abstaining from alcohol and more likely to be current drinkers and risk drinkers than women, independent of health survey, age group, and the method used to assess alcohol consumption. The gender differences in alcohol consumption were reduced over time and this were found more often when gender differences were assessed with PEth than self-reported. Consequently, health care professionals and the health authorities, as well as the public, must have awareness of the prevalence and changes in alcohol consumption among older adults in order to reduce elevated alcohol consumption among older adults to promote healthy aging.

## Supporting information

**S1 Table. Number of participants and participation rate in HUNT2 (1995–97), HUNT3 (2006–08), and HUNT4 (2017–19) surveys, by gender and age groups.**
(DOCX)

**S2 Table. Comparison of participants (≥65 years) with and without measured PEth at HUNT3 (2006–08).**
(DOCX)

**S3 Table. Comparison of participants (≥65 years) with and without measured PEth at HUNT4 (2017–19).**
(DOCX)

## Acknowledgments

We would like to acknowledge the HUNT Study participants and the Trøndelag Health Study (HUNT), which is a collaboration between HUNT Research Centre (Faculty of Medicine and Health Sciences at the Norwegian University of Science and Technology, NTNU), Trøndelag County Council, Central Norway Regional Health Authority, and the Norwegian Institute of Public Health, and the staff at the Department of Clinical Pharmacology at St. Olavs University Hospital in Trondheim (Norway) for their collaboration.

## Author Contributions

**Formal analysis:** Kjerstin Tevik, Jūratė Šaltytė Benth, Anne-Sofie Helvik.

**Methodology:** Kjerstin Tevik, Ragnhild Bergene Skråstad, Jūratė Šaltytė Benth, Geir Selbæk, Sverre Bergh, Rannveig Sakshaug Eldholm, Steinar Krokstad, Anne-Sofie Helvik.

**Writing – original draft:** Kjerstin Tevik, Ragnhild Bergene Skråstad, Jūratė Šaltytė Benth, Geir Selbæk, Sverre Bergh, Rannveig Sakshaug Eldholm, Steinar Krokstad, Anne-Sofie Helvik.

**Writing – review & editing:** Kjerstin Tevik, Ragnhild Bergene Skråstad, Jūratė Šaltytė Benth, Geir Selbæk, Sverre Bergh, Rannveig Sakshaug Eldholm, Steinar Krokstad, Anne-Sofie Helvik.

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
