## [Decision Letter · Decision Letter 0]

14 Feb 2024

PONE-D-23-36609Prevalence and change in alcohol consumption in older adults over time, assessed with self-report and Phosphatidylethanol 16:0/18:1 – The HUNT StudyPLOS ONE

Dear Dr. Tevik,

Thank you for submitting your manuscript to PLOS ONE. After careful consideration, we feel that it has merit but does not fully meet PLOS ONE’s publication criteria as it currently stands. Therefore, we invite you to submit a revised version of the manuscript that addresses the points raised during the review process.

We look forward to receiving your revised manuscript.

Kind regards,

Y-h. Taguchi, Dr. Sci.

Academic Editor

PLOS ONE

Journal Requirements:

Reviewers' comments:

Reviewer's Responses to Questions

**Comments to the Author**

1. Is the manuscript technically sound, and do the data support the conclusions?

Reviewer #1: Yes

Reviewer #2: Partly

2. Has the statistical analysis been performed appropriately and rigorously? 

Reviewer #1: Yes

Reviewer #2: Yes

3. Have the authors made all data underlying the findings in their manuscript fully available?

Reviewer #1: Yes

Reviewer #2: No

4. Is the manuscript presented in an intelligible fashion and written in standard English?

Reviewer #1: No

Reviewer #2: Yes

5. Review Comments to the Author

Reviewer #1: The manuscript reported the prevalence and change in alcohol consumption in three cohort of older adults at Norway and examined the age and gender differences. The study includes a large sample size derived from data spanning a 20-year period and incorporated both self-reported alcohol measures and the biomarker Peth. The findings contribute valuable information to the existing literature. However, the manuscript lacks conciseness, featuring repeated description across different paragraphs and excess of unnecessary overly detailed information, resulting in an overall excessive length. Major revisions are suggested before publishing.

1． Introduction: Providing a detailed introduction to the background, but it has become overly length with excessive content. It would be beneficial to focus on key points to prioritize information according to the study aims. According to my understanding, the emphasis in the background introduction should be on highlighting the significant impact of alcohol consumption on health of older adults and importance of investigating gender differences in drinking patterns.

2． The paragraph on page 9 described the differences between participants and non-participants. It’s not clear if the differences were statistically significant and how that biased the study findings.

3． Table 2 has a lot of description on how to define drinking patterns. Those should be moved to paragraphs, and the table should be simple and clear. The table can add a column for each survey to provide a brief description of its definition on these alcohol measures.

4． PEth values were categorized into 4 groups, <0.03, >0.06, >0.1, and >0.3. How about those between 0.03 and 0.06? Are they excluded from the study?

5． For table 3 and table 4, it’s not necessary to add the total numbers for each variable, which made the table very busy. It’s more informative to see the count and percentages.

6． Mean of income was reported. Since the disparity in income is substantial, sometimes it’s more informative to see categories.

7． Table 6 is a very busy table, which is challenging for the readers to follow. I suggest reformatting it for better clarity. Give its objective to compare results and changes across the three surveys, it would be suitable to present the data by surveys rather than by alcohol measures. Presenting the results from each survey side by side will make it easier to view the changes. In addition, gender differences can be presented in a separate table.

8． What’s the rationale to stratify the data by age instead of adjusting age in the model?

9． The results section described a lot of changes or differences across the three surveys. Are they all statistically significant?

10． The section title “Prevalence, change in prevalence, and gender differences in self-reported and objective measures (Peth) alcohol consumption” on page 44 is in the middle of the discussion section. It seems unnecessary to have a separate section title here.

11． Clinical implications did not mention anything related to gender disparity.

12． Some language used is not professionally precise. For example, in the sentence “Self-reported frequent drinking and risk drinking increased for both genders over time …”, need to clarify it’s the prevalence or the proportion of people with frequency drinking and risk drinking.

Reviewer #2: General comments

This is a potentially very important study with a large dataset on how alcohol consumption changes in the older age over 3 study periods. The study outline is somewhat unclear because of the overwhelming quantity of data presented, and the results are difficult to read in the current manuscript. Please see some suggestions to improve this.

Abstract

The study period seems to be 24 years, not 20.

Introduction

The introduction is too long and too broad for the more limited research aim of the study. It should be more focused on alcohol consumption among older and changing patterns in older adults. These topics are covered but the rest of the text can be more concise.

There are 98 references in the intro, the authors must prioritize the most relevant references rather than having up to 10+ references for one statement.

Methods

A description of how the participants who were analysed for PEth was selected, seems to be missing. Described on P 13 line 254-257.

Page 9 second paragraph contains study results which should be reported in the results section.

P 12 line 226, why is Statistics Norway a headline?

P 13 and table 2. To understand the study it is important to have a clear description of how alcohol consumption was recorded thru the questionnaires and how the cutoff values was defined in the blood samples. None of this is very clear in the manuscript.

First give a clear overview of the difference in alcohol questions between the three different samples. This can be done in a table.

The further detailed descriptions must be moved from the table to the text, and be clarified.

The different cut-off levels of PEth must be justified with some references to earlier studies for each value.

Results

Start with information on study participation etc. partly described in methods.

The cohort effect is not described e.g. that the participants in Hunt 2 will be ten years older in Hunt 3 and 20 years older in Hunt 4. It would be interesting to see the development compared thru the increasing age groups rather than compare the same age groups.

Table 3 should include self-reported alcohol and PEth results.

Table 4 and 5 provides little relevant information.

Table 6 and 7 must be simplified, it is very difficult to understand the data presented. The aim is to examine the prevalence and changes in the alcohol consumption. A figure might be a better way to show a changing intake.

Overall the result section has to be re- constructed in order to give a more clear presentation to the reader. The tables presented are very difficult to read and it is unclear why the data collections are analysed and presented separately when then aim is to present changes between the three studies.

The change over time both in crude numbers and adjusted should be presented in one table or figure and the authors have to choose which variables are relevant to compare with. E.g. both income and after tax income and urban/ rural living are co- variates. What is the rationale for including these variables? It is stated several times that there are no large cities in the area, a definition on “rural/urban” is needed.

The way the results are presented now it is difficult to see the changes, as the findings are presented separately.

Overall try to show the changes in PEth and self-reported consumption thru the observation period, and focus on the aim of the study when presenting the data.

Discussion

Overall, focusing more on the overall changes than the point estimates would benefit the discussion.

Comparisons between low PEth values (< 0.030) to lifetime abstaining or low alcohol consumption is problematic. As there is little overlap in the observation period (lifetime vs. 2-3 weeks for PEth).

Interactions with medicinal drugs should be mentioned in the clinical implications section.

6. PLOS authors have the option to publish the peer review history of their article (what does this mean?). If published, this will include your full peer review and any attached files.

Reviewer #1: No

Reviewer #2: No

---

## [Author Response · Author response to Decision Letter 0]

25 Mar 2024

PONE-D-23-36609

Prevalence and change in alcohol consumption in older adults over time, assessed with self-report and Phosphatidylethanol 16:0/18:1 – The HUNT Study

PLOS ONE

PONE-D-23-36609

Prevalence and change in alcohol consumption in older adults over time, assessed with self-report and Phosphatidylethanol 16:0/18:1 – The HUNT Study

PLOS ONE

1. We note that the grant information you provided in the ‘Funding Information’ and ‘Financial Disclosure’ sections do not match. 

In the ‘Funding Information’ we have added that Ragnhild Bergene Skråstad received the funding from St. Olavs University Hospital (Trondheim, Norway) (no grant number) and from the Norwegian DAM Foundation in cooperation with the Norwegian non-profit organization ‘Av og til’ (no grant number).

2. Have the authors made all data underlying the findings in their manuscript fully available?

Reviewer #1: Yes

Reviewer #2: No

Due to restrictions imposed by the HUNT Research Centre (in accordance with the Norwegian Data Inspectorate), data cannot be made publicly available. Data are currently stored in the HUNT databank, and there are restrictions in place for the handling of HUNT data files. Data used from the HUNT Study in research projects will be made available on request to the HUNT Data Access Committee (kontakt@hunt.ntnu.no). The HUNT data access information (available here: http://www.ntnu.edu/ hunt/data) describes in detail the policy regarding data availability.

Review Comments to the Author

Reviewer #1: The manuscript reported the prevalence and change in alcohol consumption in three cohort of older adults at Norway and examined the age and gender differences. The study includes a large sample size derived from data spanning a 20-year period and incorporated both self-reported alcohol measures and the biomarker Peth. The findings contribute valuable information to the existing literature. However, the manuscript lacks conciseness, featuring repeated description across different paragraphs and excess of unnecessary overly detailed information, resulting in an overall excessive length. Major revisions are suggested before publishing.

We would like to thank the reviewer for their contribution for making the article better. When referring to page and line number in the response of the review, please see the Revised Manuscript with Track Changes. 

1． Introduction: Providing a detailed introduction to the background, but it has become overly length with excessive content. It would be beneficial to focus on key points to prioritize information according to the study aims. According to my understanding, the emphasis in the background introduction should be on highlighting the significant impact of alcohol consumption on health of older adults and importance of investigating gender differences in drinking patterns.

We agree that the introduction is too detailed resulting in an excessive length. The introduction is shortened, and the focus is now on changes in alcohol consumption among older adults, impact of alcohol consumption on older adults’ health, and gender differences. 

2． The paragraph on page 9 described the differences between participants and non-participants. It’s not clear if the differences were statistically significant and how that biased the study findings.

The differences between participants and non-participants in HUNT3, described on page 9 from line 217, are statistically significant. The results can be found in Additional file 1 and 2 in Langhammer et al. 2012 (https://link.springer.com/article/10.1186/1471-2288-12-143). The differences between participants and non-participants in HUNT4 were not assessed for statistical significance (page 10, from line 223), and thus not available. For further information see Supplementary Table S3 in Åsvold et al. 2023 (Cohort Profile Update: The HUNT Study, Norway | International Journal of Epidemiology | Oxford Academic (oup.com). How the differences between participants and non-participants may have biased the study findings are described in the limitation section on page 80 from line 974. 

3． Table 2 has a lot of description on how to define drinking patterns. Those should be moved to paragraphs, and the table should be simple and clear. The table can add a column for each survey to provide a brief description of its definition on these alcohol measures.

We appreciate this suggestion, and the description on how to define the drinking patterns are moved to the text. Table 2 (page 20, line 356) is made simpler, and we have added a column for each HUNT survey to provide a brief definition of the different alcohol measures.

4． PEth values were categorized into 4 groups, <0.03, >0.06, >0.1, and >0.3. How about those between 0.03 and 0.06? Are they excluded from the study?

Participants with PEth concentrations between 0.03 and 0.06 µmol/l were not excluded from the study. In the method section under the headline “Statistics” (page 28, line 396) it is described that alcohol consumption assessed by PEth is dichotomized as <0.03 vs. ≥0.03, >0.06 vs. ≤0.06, >0.10 vs. ≤0.10, and >0.30 vs. ≤0.30 µmol/l. Thus, participants with PEth values between 0.03 and 0.06 µmol/l were included in the group of participants with either PEth concentrations ≥0.03, ≤0.06, ≤0.10, or ≤0.30 µmol/l. The PEth concentrations were dichotomized in this way (<0.03 vs. ≥0.03, >0.06 vs. ≤0.06, >0.10 vs. ≤0.10, and >0.30 vs. ≤0.30 µmol/l) in order to use the results from the HUNT-study by Skråstad et al. 2023 (Skråstad et al., 2023) where they quantified alcohol consumption in the general population by analyzing Phosphatidylethanol. Thus, we do not have data about the proportion of participants with PEth concentrations between 0.03 and 0.06 µmol/l.

5． For table 3 and table 4, it’s not necessary to add the total numbers for each variable, which made the table very busy. It’s more informative to see the count and percentages.

We have chosen to keep the data presented in Table 3 (page 30, line 431). However, we are not quite sure whether reviewer means Table 3 and original Table 4, or original Table 4 and original Table 5. Nevertheless, original Table 4 and Table 5 are moved to the Supplemental section (S2 Table and S3 Table) as another reviewer characterized these tables as little informative. 

6． Mean of income was reported. Since the disparity in income is substantial, sometimes it’s more informative to see categories.

Categorization of a continuous variable implies a considerable reduction in information. For this reason, we would like to avoid it. However, we agree that a more detailed description of the income variable is appropriate to provide. We therefore include median and first and third quartiles in addition to the mean into the descriptive table (Table 3), and hope the reviewer finds this satisfactory.

7． Table 6 is a very busy table, which is challenging for the readers to follow. I suggest reformatting it for better clarity. Give its objective to compare results and changes across the three surveys, it would be suitable to present the data by surveys rather than by alcohol measures. Presenting the results from each survey side by side will make it easier to view the changes. In addition, gender differences can be presented in a separate table.

We agree that original Table 6 (now Table 4, page 38) was very busy. The format of the table is changed, and the data is presented by the three surveys rather than by alcohol measures. The gender differences are presented in a separate table (see page 47, Table 6). We think these changes give the table more clarity and make the table suitable.

8． What’s the rationale to stratify the data by age instead of adjusting age in the model?

Previous studies have shown that alcohol consumption decreased by increasing age group (Bratberg et al., 2016; Immonen et al., 2011; Stelander et al., 2021). Thus, we wanted to stratify the data by age in order to demonstrate potential differences in age. This would not be possible by including age as adjustment variable in the regression model. Moreover, we also wanted to examine whether the gender differences differed between age groups.

9． The results section described a lot of changes or differences across the three surveys. Are they all statistically significant?

Overall changes in different drinking patterns across the three surveys are not assessed for statistical significance. We presented overall changes in alcohol consumption descriptively by using the results in Table 4 (page 38) and Table 5 (page 43). In the method section we have added the following on page 28 from line 409:

“Overall changes in alcohol consumption assessed with self-report and PEth across the surveys were presented only descriptively”.

The gender differences within each HUNT survey and across the three HUNT surveys were examined by a logistic regression model and presented as odds ratios with corresponding 95% confidence interval. Only statistically significant adjusted results regarding gender differences are presented in the Results section, and in the Method section we have added “statistically significant” to clarify this for the reader (page 28, line 407): 

“Only statistically significant adjusted results regarding gender differences within each HUNT survey and across the HUNT surveys are presented”.

10． The section title “Prevalence, change in prevalence, and gender differences in self-reported and objective measures (Peth) alcohol consumption” on page 44 is in the middle of the discussion section. It seems unnecessary to have a separate section title here.

We agree, and the title is removed (page 71, line 763). 

11． Clinical implications did not mention anything related to gender disparity.

Thanks for pointing out this lack. Now, the gender disparity is added to Clinical implications on page 78 from line 934:

“The gender convergence in alcohol consumption among older adults with women moving towards men’s drinking patterns, highlights the need for health care professionals to inform older women about their increased risk of adverse effects of alcohol use”.

12． Some language used is not professionally precise. For example, in the sentence “Self-reported frequent drinking and risk drinking increased for both genders over time …”, need to clarify it’s the prevalence or the proportion of people with frequency drinking and risk drinking.

Thank you for making us aware of this unprecise sentence which we have clarified (page 71, line 753):

“…the prevalence of self-reported frequent drinking and risk drinking increased for both genders over time”.

Reviewer #2: General comments

This is a potentially very important study with a large dataset on how alcohol consumption changes in the older age over 3 study periods. The study outline is somewhat unclear because of the overwhelming quantity of data presented, and the results are difficult to read in the current manuscript. Please see some suggestions to improve this.

We would like to thank the reviewer for their contribution for making the article better. When referring to page and line number in the response of the review, please see the Revised Manuscript with Track Changes. 

Abstract

1. The study period seems to be 24 years, not 20.

The study period is changed from 20 to 24 years in the abstract (page 3, line 54 and page 4, line 78). 

2. Introduction

The introduction is too long and too broad for the more limited research aim of the study. It should be more focused on alcohol consumption among older and changing patterns in older adults. These topics are covered but the rest of the text can be more concise.

There are 98 references in the intro, the authors must prioritize the most relevant references rather than having up to 10+ references for one statement.

We agree that the introduction is too detailed and too long. The introduction is shortened, and we have kept the focus on changing drinking patterns in older adults, health consequences of alcohol consumption, and gender differences in older adults. The number of references in the introduction is reduced from 98 references to 47 references. 

Methods

1. A description of how the participants who were analysed for PEth was selected, seems to be missing. Described on P 13 line 254-257.

We understand the relevance of this request. How participants were selected for PEth is included on page 14, from line 264: 

“In HUNT3, PEth was analyzed in stored blood from the HUNT-biobank and material was only available from a subsample. For practical reasons in HUNT4, the collection of blood samples for PEth-analysis did not start until approximately halfway through the study period, and thus PEth-analysis is only available in a subsample. In both cases, the subsample was independent of whether the participants reported to have a high or a low level of alcohol consumption”,

2. Page 9 second paragraph contains study results which should be reported in the results section.

The results presented on page 9, second paragraph (now third paragraph, line 213), are previously published information (Holmen et al., 2003; Krokstad et al., 2013; Langhammer et al., 2012; Åsvold et al., 2022). They are partly based on the general population in the HUNT surveys, and not only restricted to older adults (≥65 years). The information was reported to give information about participation rates and transparency about the data. We have chosen to keep these results in the Method section. However, to reduce the total number of tables in the main manuscript, original Table 1, presenting the results regarding the participation rate among those aged 60 years and older in HUNT2, HUNT3, and HUNT4, is moved to the Supplemental section (S1 Table).

3. P 12 line 226, why is Statistics Norway a headline?

The headline “Statistic Norway” is removed (page 10, line 232)

4. P 13 and table 2. To understand the study it is important to have a clear description of how alcohol consumption was recorded thru the questionnaires and how the cutoff values was defined in the blood samples. None of this is very clear in the manuscript.

First give a clear overview of the difference in alcohol questions between the three different samples. This can be done in a table.

The further detailed descriptions must be moved from the table to the text, and be clarified.

The different cut-off levels of PEth must be justified with some references to earlier studies for each value.

A detailed description of the different questions used in HUNT2, HUNT3, and HUNT4 to measure alcohol consumption is found in Table 1 (page 15). In the initial Manuscript this table was added as a Supplemental Table. 

The definitions of the different drinking patterns are removed from Table 2 (page 20) and added to the text. Some details about the alcohol questions are added to the text (see page 17 from line 290). Reference(s) are added to each PEth value (page 19, from line 352). Table 2 (page 20) is kept, but simplified, and contains now just the definition of the drinking patterns in HUNT2, HUNT3 and HUNT4 and the stratified PEth concentrations.

5. Results

Start with information on study participation etc. partly described in methods.

As previously mentioned, we have chosen to keep the information about the study participation in the method section as these results are published in previous studies (Holmen et al., 2003; Krokstad et al., 2013; Åsvold et al., 2022).

The cohort effect is not described e.g. that the participants in Hunt 2 will be ten years older in Hunt 3 and 20 years older in Hunt 4. It would be interesting to see the development compared thru the increasing age groups rather than compare the same age groups.

We fully agree, it is interesting to follow the participants longitudinally by increasing age from HUNT2 and further to HUNT3 and HUNT4. However, we have chosen to conduct a separate study to see the development of different drinking patterns by increasing age. This study assessing the participants longitudinally is ongoing. 

Table 3 should include self-

---

## [Decision Letter · Decision Letter 1]

17 May 2024

Prevalence and change in alcohol consumption in older adults over time, assessed with self-report and Phosphatidylethanol 16:0/18:1 – The HUNT Study

PONE-D-23-36609R1

Dear Dr. Tevik,

We’re pleased to inform you that your manuscript has been judged scientifically suitable for publication and will be formally accepted for publication once it meets all outstanding technical requirements.

Kind regards,

Y-h. Taguchi, Dr. Sci.

Academic Editor

PLOS ONE

Additional Editor Comments (optional):

Reviewers' comments:

Reviewer's Responses to Questions

**Comments to the Author**

1. If the authors have adequately addressed your comments raised in a previous round of review and you feel that this manuscript is now acceptable for publication, you may indicate that here to bypass the “Comments to the Author” section, enter your conflict of interest statement in the “Confidential to Editor” section, and submit your "Accept" recommendation.

Reviewer #2: All comments have been addressed

2. Is the manuscript technically sound, and do the data support the conclusions?

Reviewer #2: Yes

3. Has the statistical analysis been performed appropriately and rigorously? 

Reviewer #2: Yes

4. Have the authors made all data underlying the findings in their manuscript fully available?

Reviewer #2: No

5. Is the manuscript presented in an intelligible fashion and written in standard English?

Reviewer #2: Yes

6. Review Comments to the Author

Reviewer #2: The manuscript has been much improved.

I suggest to change one formatting issue:

In the subsection Study setting, data sources and participants - please include the subsection Participants rather than having a separate for participants (p 8- line 188)

7. PLOS authors have the option to publish the peer review history of their article (what does this mean?). If published, this will include your full peer review and any attached files.

Reviewer #2: No

---

## [Editor Report · Acceptance letter]

22 May 2024

PONE-D-23-36609R1 

PLOS ONE

Dear Dr. Tevik, 

I'm pleased to inform you that your manuscript has been deemed suitable for publication in PLOS ONE. Congratulations! Your manuscript is now being handed over to our production team.

Kind regards, 

on behalf of

Professor Y-h. Taguchi 

Academic Editor

PLOS ONE